# Traditional Logic and Computational Thinking

**J.-Martín Castro-Manzano** 

Faculty of Philosophy, UPAEP University, Puebla 72410, Mexico; josemartin.castro@upaep.mx

**Abstract:** In this contribution, we try to show that traditional Aristotelian logic can be useful (in a non-trivial way) for computational thinking. To achieve this objective, we argue in favor of two statements: (i) that traditional logic is not classical and (ii) that logic programming emanating from traditional logic is not classical logic programming.

**Keywords:** syllogistic; aristotelian logic; logic programming

## 1. Introduction

Computational thinking, like critical thinking [1], is a sort of general-purpose thinking that includes a set of logical skills. Hence logic, as a scientific enterprise, is an integral part of it. However, unlike critical thinking, when one checks what "logic" means in the computational context, one cannot help but be surprised to find out that it refers primarily, and sometimes even exclusively, to classical logic (cf. [2–7]).

Classical logic is the logic of Boolean operators, the logic of digital circuits, the logic of Karnaugh maps, the logic of Prolog. It is the logic that results from discarding the traditional, Aristotelian logic in order to favor the Fregean–Tarskian paradigm. Classical logic is the logic that defines the standards we typically assume when we teach, research, or apply logic: it is the received view of logic.

This state of affairs, of course, has an explanation: classical logic works wonders! However, this is not, by any means, enough warrant to justify the absence or the abandon of traditional logic within the computational thinking literature. And so, in this contribution, we try to take advantage of this educational gap in order to show that traditional logic can be useful (in a non-trivial way) for computational thinking.

To achieve this goal, we argue in favor of two claims: (i) that traditional logic is not classical logic (and hence it will not suffice to claim that traditional logic is already included in classical logic), and (ii) that logic programming emerging from traditional logic is not classical logic programming (and thus it will not be enough to assume that the logic programming paradigm already includes this sort of programming as is).

We have organized this paper in the following way. First, we make explicit the received view of logic and logic programming; then, we argue that traditional logic is not classical logic, and with this result, we show what traditional logic programming would look like. At the end, we comment on how these claims help close said educational gap. We hope to show yet another link between philosophy and computation.

## 2. The Received View of Logic (Programming)

Broadly speaking, the *raison d'être* of logic is the study of inference within natural language [8,9], and in order to study inference in this sense it is customary to use classical logic, a logic defined by first-order languages.

The origin of this habit has an interesting history [10] related to the representational advantages first order languages offer over traditional systems—today, it is commonplace to mention how [11–15] contributed to this custom; however, even if this syntactic standard—that of using first order systems—is common to us when teaching, researching, or applying logic—after all, this is the received view of logic—there is no need to be particularly acute

in order to notice that this view of logic may indeed be familiar, but that does not make it natural. Woods comments:

> *"It is no secret that classical logic and its mainstream variants are not much good for human inference as it actually plays out in the conditions of real life—in life on the ground, so to speak. It is not surprising. Human reasoning is not what the modern orthodox logics were meant for. The logics of Frege and Whitehead & Russell were purpose-built for the pacification of philosophical perturbation in the foundations of mathematics, notably but not limited to the troubles occasioned by the paradox of sets in their application to transfinite arithmetic."* ([16], p. 404)

It is almost a truism that classical logic has been instrumental for the study of inference in general, but it does not cease to surprise us that, despite its original purpose in the foundations of mathematics, it is constantly used as the *bona fide* tool for representing inference in natural language, as Englebretsen might say [17].

Certainly, this should not be a revelation because we know classical logic "has been developed primarily by mathematicians with mathematical applications in mind" ([18], p. 4) but the issue is that, as Kreeft would put it, logic was made for us writ large, not inversely ([19], p. 23). Of course, we are not claiming classical logic and its exploits are in the path of doom. Quite the contrary. But in our view, if logic is about inference, logic needs not abandon traditional systems.

And it has not. For one, and closer to Aristotle than to Frege, Fred Sommers championed a revision of traditional logic under the assumption that logic has to deal primarily with natural language. Since the late 60's, his project unfolded into three projects on ontology, semantics, and logic (cf. [20]) that became, respectively, a theory of categories, a theory of truth, and a theory of logic known as Term Functor Logic [17,21–25], a plus-minus algebra that uses terms—in an Aristotelian fashion—rather than first-order language elements such as variables or quantifiers.

We will return to this particular logic later, but what we want to stress is that there has been a mismatch between logic and natural language that has led to an overestimation of classical logic. And so, there is some evidence to the effect that the cultivation of Aristotelian logic, in spite of certain current efforts [9,17,19,25–29], has been disparaged in various ways, especially since the early 20th century.

However, this story does not end here. This received view of logic has a computational counterpart because there is also a received view of logic programming. Indeed, when, for example, ([30], p. 3) mention that "logic" has been used as a tool for designing computers and for reasoning about programs and logic programming, they are talking about classical logic again. And the same happens when one reviews the foundational or the usual literature on logic programming [31–34]. However, as we shall see, traditional logic is not classical and, therefore, the logic programming derived from it might not be classical or, better yet, needs not be classical. But let us not get ahead of ourselves.

Arguably, the firsts attempts to relate (classical) logic and programming can be traced back to Church [35–37] and Turing [38], although the patent results of these two projects can be better appreciated in two main areas: knowledge representation and programming. And so for McCarthy, one of the founding fathers of artificial intelligence, classical first-order logic was his weapon of choice [39]; whereas Kowalski and Colmerauer developed Prolog, the first logic programming language, out of first-order logic [31,32].

Of course, both areas soon discovered the need to include non-classical augmentations, such as non-monotonicity or negation as failure, but the Fregean–Tarskian imprint has been there by design. And hence in the famous Kolwalski's equation [40], *Algorithm = logic + control*, "logic" means, essentially, classical logic.

## 3. Classical vs. Traditional Logic

Now, there is the common assumption that traditional logic, namely syllogistic, is classical logic. This is not a fringe belief: some popular handbooks, for example, make this assumption explicit ([41], p. 168); while some authoritative scholars, for instance, make

it implicit ([8], p. 5). However, we are afraid this presumption does not make justice to neither party: it commits the traditional logicians to the claim that their logic is a sub-logic of classical logic, and it commits the classical logicians to the claim that their logic is a super-logic of traditional logic.

These commitments, however, imply a condition that does not seem to hold. Consider that if these commitments were true, we would expect some sort of continuity from traditional logic to classical logic, but as we will argue, this is not the case: there is a subtle but crucial difference between being a sub-logic of classical logic and being a sub-classical logic. It looks as if traditional logic is sub-classical, but not a sub-logic of classical logic.

Although there are several ways to characterize classical logic, for the purposes of this paper, we consider a logic to be classical when it drops the traditional ternary syntax (subject-copula-predicate) to promote the binary syntax (function-argument); when it admits the explosion principle or *ex falso* (i.e., $p \wedge \neg p \vdash q$, for any $q$, that is to say, everything follows from falsehood); when it admits the positive paradox of implication or *verum ad* (i.e., $q \vdash p \vee \neg p$, for any $q$, that is to say, a truth follows from everything); and when it holds the reflexivity of the inference relation (i.e., $p \vdash p$) as if it were the same as the identity principle (i.e., $\vdash p \Rightarrow p$).

Clearly, the logic we have called "classical" is classical in this sense, but syllogistic, the core of traditional logic is not. Formally, syllogistic is a term logic that deals with inference between categorical statements (*vide* Appendix A), and from a larger point of view, it is also an integral part of what we could call a basic *corpus aristotelicum* that, in turn, could be defined by the tuple $\mathfrak{A} = \langle B_E, B_C, B_O, B_P, B_L \rangle$ (cf. [42], p. 4ff) where $B_E$ is an epistemological theory that includes the production of hypothesis and inferences under the Aristotelian notions of *epagogé* (i.e., induction) and *syllogismós* (i.e., deduction), respectively; $B_C$ is a theory of causality that distinguishes material, formal, efficient, and final causes; $B_O$ is an ontological theory that assumes a systemic view of the world given the double claim that there are no unhad properties (*contra* universals *ante rem*) nor objects without properties (*contra* bare particulars); $B_P$ is a psychological theory that makes use of the concept of "habit" in order to explain behavior (both *epagogé* and *syllogismós*, for example, would be habits when performed by agents); and $B_L$ is a logical theory designed for dealing with categories, statements, inferences, explanations, argumentation, and inferential mistakes, that includes syllogistic as a theory of deductive inference particularly crafted to avoid irrelevance, as explained by Thom's display of Kilwardby's first exposition of syllogistic—also called the Boethian exposition (Figure 1).

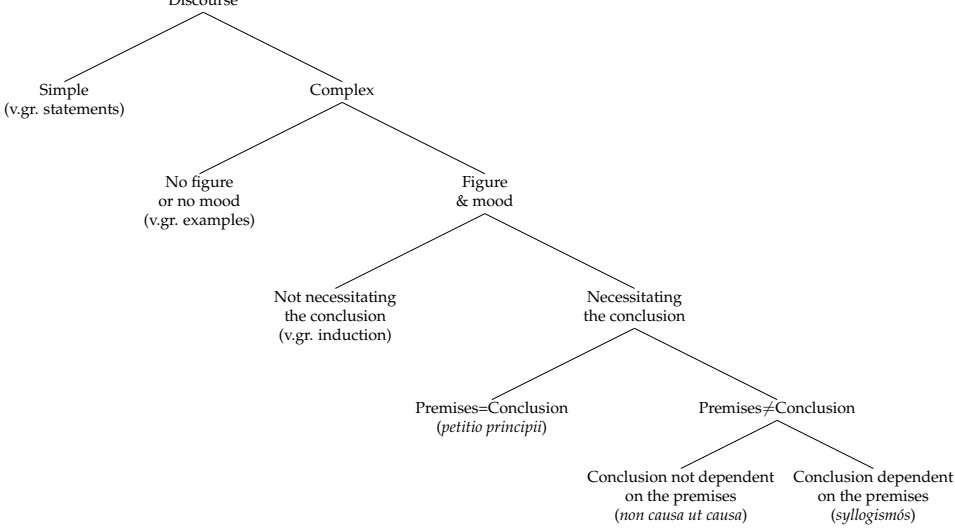

**Figure 1.** The Boethian exposition (adapted from ([43], p. 44).

According to the Boethian exposition, a syllogistic inference or syllogism—*syllogismós*—is a piece of complex discourse (in so far as it includes at least two premises and one conclusion) with mood and figure (because the order of statements and terms matters) in which a conclusion that is different from the premises (thus avoiding *petitio principii*, a fallacy also known as *begging the question*) necessarily (and hence deductively) follows from and dependes on said premises (thus avoiding irrelevance, *non causa ut causa*).

This account, based upon a commentary to the *Prior Analytics* [44], is not outdated nor marginal. A similar yet independent assessment has been given by [45], but based upon Aristotle's earlier logic, namely the *Topics*, *On Sophistical Refutations* and *On Interpretation*. According to this other study, syllogistic inference complies, among others, with the following properties:

- Minimality: Syllogistic inferences are minimal in so far as they contain the premises needed for their validity and none other.
- Non-Circularity: Syllogistic inferences are elementarily non-circular, that is, their conclusions repeat no premises.
- Premise multiplicity: Syllogistic inferences are multi-premised.
- Premise consistency: Syllogistic inferences admit only consistent premises.
- The because-of condition: Syllogistic inference is valid if it excludes terms from the outside; only if, that is, each term in its conclusion has an occurrence in at least one premise and every premise has a term occurring in the conclusion.

Thus, according to Woods' exposition, a syllogism is a (finitely premised) argument that satisfies minimality, non-circularity, premise multiplicity, premise consistency and the because-of condition. And this assessment initially suggests—although such suggestion is later discussed—that syllogistic inference is relevant, paraconsistent, and non-monotonic in so far as it behaves as a linear logic, excludes explosion, and is minimal. And these features, properly organized, avoid *ex falso* and *verum ad*.

We can try to accommodate both assessments, Boethius' and Woods', in order to suggest that there is no continuity from traditional to classical logic, in spite of popular opinion. Ponder, thus, Figure 2, and notice, hence, that it will not suffice to claim that traditional logic is already included in classical logic. And recall, additionally, that traditional logic uses a term syntax, not a Fregean syntax; that the notion of inference in classical logic requires reflexivity and monotonicity but does not make use of premise multiplicity, whereas traditional logic requires premise multiplicity but avoids reflexivity or monotonicity. And so, all things considered, these remarks should probably be enough to distinguish traditional from classical logic, or at least enough to reject a smooth continuity from traditional to classical logic.

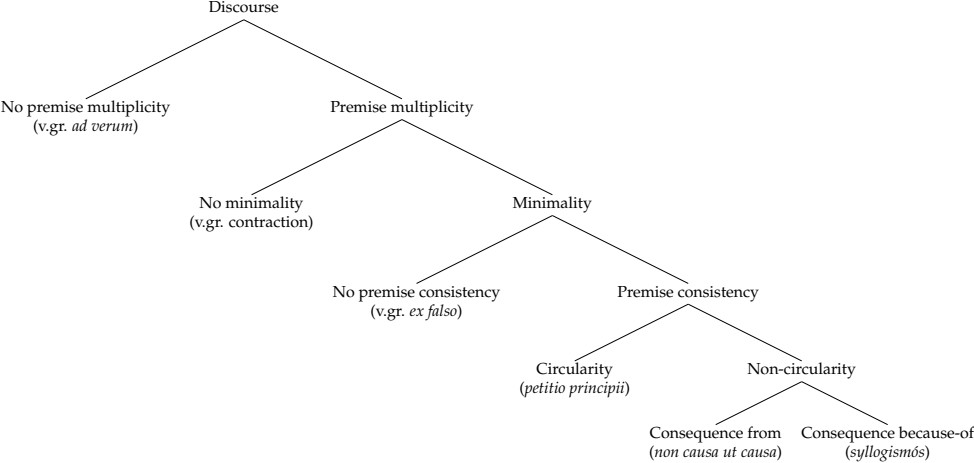

**Figure 2.** Woods' exposition.



## 4. Traditional Logic Programming vs. Classical Logic Programming

Let us now return to programming. Consider that Mozes has already argued that the abandon of traditional logic in the field of computer science is not justified [46]. In his opinion, traditional logic was originally created in order to understand reasoning in natural language, and so traditional logic not only has philosophical or cognitive import, but computational relevance. Consequently, Mozes developed the concept of Aristotelian database.

A database is Aristotelian when it complies certain abilities and structures. So, among others, a database is Aristotelian when it has the ability to provide explanations in natural language; when it can offer information, in response to dichotomous questions ("yes/no"), when a stronger or weaker versions of a "yes" can be tested; when it has the ability to point out results that cannot be proven but are possible; when it can suggest implicit rules that, if added to the database, could provide affirmative answers; and when it can indicate instances in which non-deductive patterns, such as analogies, may be helpful.

Structurally, we can divide the representative and inferential aspects of an Aristotelian database. From the standpoint of knowledge representation, it consists of a set of constants that stands for objects and a set of relations—there are no functions—to stand for properties of objects. So, for example, `human(Socrates)` (i.e., *Socrates is a human being*) is a *fact*, whereas a *rule*, on the other hand, consists of a subject, which is the conjunction of one or more relations applied to variables and constants, and a predicate, which is a unique relation, plus a type of rule that indicates the connection between the subject and the predicate. So, when one writes a rule the predicate goes first, then the type of the rule, and finally the subject, for example, `mortal(X) A human(X)` means *Every human being is mortal*.

It is clear, then, these bases use a syntax similar to that of (pure) Prolog, and Prolog, like classical logic, is the logic programming language *par excellence*. In Prolog, a program is a set of statements defined by two kinds of (Horn) clauses: facts and rules. A rule has the form `Head : −Body` where `Head` and `Body` are first-order statements (and a fact is a clause with an empty body). But more importantly, from the standpoint of inference, Prolog uses resolution over facts and rules; whereas Aristotelian databases, by contrast, use syllogistic as their main inference model.

Now, by following Mozes' proposal, but different from Massie's patent [47], in other place we have presented some sort of logic programming with terms [48], but unlike Mozes, who still uses a Prolog-like syntax, we use the aforementioned term logic developed by Sommers and Englebretsen (*vide* Appendix B) to obtain the following grammar:

```
<program> ::= <statement><statement>|<statement><program>
<statement> ::= <term><term>
<term> ::= <+T>|<-T>|<+t>|<-t>
```

This syntax yields a programming language in which programs look like sets of multi-premised categorical statements *à la* Sommers and Englebretsen, for example, as follows:

```
-s+H
-H+A
-H+O
-A+O
```

where `s` stands for "Socrates", `M` for "human being", `A` for "animal", and `O` for "mortal", so that this program states *Socrates is a human being*, *Human beings are animals*, *Human beings are mortal*, and *Animals are mortal*. And if we now perform the following query ("> s" stands for "What about—in this case—Socrates?"), we will notice that the answers are given syllogistically as follows ("-----" separates premises from conclusions):

```
> s
-H+A
-s+H
-----
```

```
-s+A

-H+O
-s+H
-----
-s+O
```

Given these developments, we can see some clear differences between this type of programming and classical logic programming, as in Table 1. For a start, from the knowledge representation standpoint, there is a clear syntactical difference. In traditional logic programming the binary, Fregean syntax is abandoned in order to return to the traditional term syntax: we do not use variables, constants, relations, or connectives, but rather terms and functors (hence, in traditional logic programming the distinction between facts and rules disappears).

**Table 1.** A short comparison.

| English | Classical Logic Notation | Prolog Notation | Term Logic Notation | Traditional Programming |
|---|---|---|---|---|
| Every logician is mad. | $\forall x(Lx \Rightarrow Mx)$ | m(X):-l(X). | $-L+M$ | $-L+M$ |
| No logician is mad. | $\forall x(Lx \Rightarrow \neg Mx)$ | m(X):-\ +l(X). | $-L-M$ | $-L-M$ |
| Some logician is mad. | $\exists x(Lx \wedge Mx)$ | l(a). | $+L+M$ | $+L+M$ |
|  |  | m(a). |  |  |
| Some logician is not mad. | $\exists x(Lx \wedge \neg Mx)$ | l(a). | $+L-M$ | $+L-M$ |

And from the point of view of inference, traditional logic programming performs inference by syllogistic, not exactly by resolution (cf. [49]), and so its inference engine is related to natural language by design, rather than mathematical inference. This is a crucial difference because it shows the logic behind logic programming need not be classical, and so traditional logic programs, purposely, are complex, minimal, consistent, non-circular, explanatory programs, just like syllogisms writ large. We can summarize this state of affairs in Figure 3.

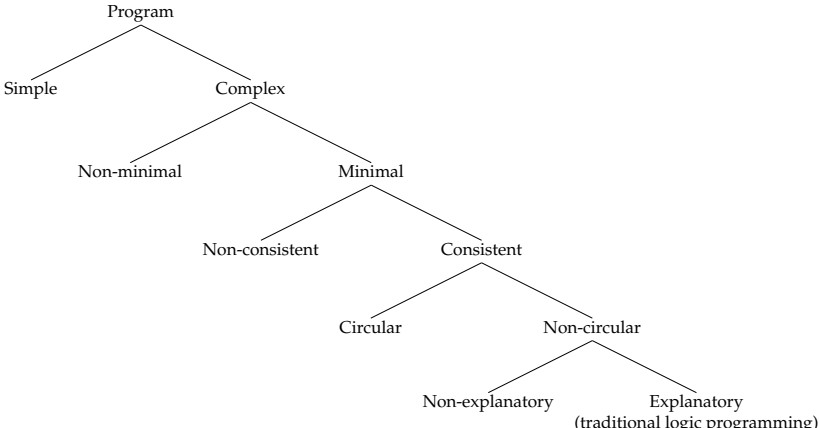

**Figure 3.** Traditional logic programming.

## 5. Conclusions

In this work, we have recovered some ideas of the traditional, Aristotelian logic in order to observe some of its qualities in the context of computational thinking. More precisely, we have tried to take advantage of an educational gap within the typical computational thinking literature—namely, that traditional logic is missing—in order to suggest that good old logic can be useful (in a non-trivial way) for computational thinking.

To achieve this goal, we have argued in favor of two claims: that traditional logic is not classical and that the logic programming emanating from traditional logic is not classical

logic programming. These claims support the notions that it will not suffice to claim that traditional logic is already included in classical logic, and that it will not be enough to assume that the logic programming paradigm already includes this sort of traditional programming as is. And therefore, these notions are not trivial, and thus not talking about them, in the computational thinking literature, does not seem justified.

With the consecution of this goal, we hope we have shown yet another link between philosophy and computation. In short, we could say that, in the context of computational thinking, we do not pretend to go back to Aristotle full force, by any stretch of the imagination, but we do not see genuine reasons to forget him, and we forget him at our peril.

**Funding:** This research was funded by an UPAEP Research Grant.

**Institutional Review Board Statement:** Not applicable.

**Informed Consent Statement:** Not applicable.

**Data Availability Statement:** Not applicable.

**Acknowledgments:** We would like to thank the referees for valuable comments and suggestions.

**Conflicts of Interest:** The author declares no conflict of interest.

### Appendix A. Syllogistic

*Syllogistic* is a term logic that deals with inference between categorical statements. A *categorical statement* is a statement composed by two terms, a quantity, and a quality. The subject and the predicate of a statement are called *terms*: the term-schema S denotes the subject term of the statement and the term-schema P denotes the predicate. The *quantity* may be either universal (*All*) or particular (*Some*) and the *quality* may be either affirmative (*is*) or negative (*is not*). These categorical statements have a *type* denoted by a label (either a (universal affirmative, SaP), e (universal negative, SeP), i (particular affirmative, SiP), or o (particular negative, SoP)) that allows us to determine a *mood*, that is, a sequence of three categorical statements ordered in such a way that two statements are premises (major and minor) and the last one is a conclusion. A *categorical syllogism*, then, is a mood with three terms one of which appears in both premises but not in the conclusion. This particular term, usually denoted with the term-schema M, works as a link between the remaining terms and is known as the middle term. According to the position of this middle term, four *figures* can be set up in order to encode the valid syllogistic moods. For the sake of brevity, but without loss of generality, here we omit the syllogisms that require existential import.

**Table A1.** Valid syllogistic moods by figure.

| First | Second | Third | Fourth |
|-------|--------|-------|--------|
| aaa | eae | iai | aee |
| eae | aee | aii | iai |
| aii | eio | oao | eio |
| eio | aoo | eio | |

### Appendix B. Term Functor Logic

*Term Functor Logic* (TFL, for short) [17,22–25] is a plus-minus algebra that employs terms and functors rather than first order language elements such as individual variables or quantifiers (cf. [9,20,22,50–52]). According to this algebra, the four categorical statements can be represented by the following syntax [17]:

a.　$SaP := -S + P$
b.　$SeP := -S - P$
c.　$SiP := +S + P$
d.　$SoP := +S - P$

Given this representation, TFL provides a simple rule for syllogistic inference: a conclusion follows validly from a set of premises if, and only if , *(i)* the sum of the premises is algebraically equal to the conclusion and *(ii)* the number of conclusions with particular quantity (viz., zero or one) is the same as the number of premises with particular quantity ([17], p. 167). Thus, for instance, if we consider a valid syllogism (say, a syllogism aaa of the first figure, aaa-1), we can see how the application of this rule produces the right conclusion (Table A2).

**Table A2.** A valid syllogism.

| | Statement | TFL |
|---|---|---|
| 1. | All computer scientists are animals. | $-C + A$ |
| 2. | All logicians are computer scientists. | $-L + C$ |
| $\vdash$ | All logicians are animals. | $-L + A$ |

In this example, we can clearly see how the rule works: *i)* if we add up the premises, we obtain the algebraic expression $(-C + A) + (-L + C) = -C + A - L + C = -L + A$, so that the sum of the premises is algebraically equal to the conclusion and the conclusion is $-L + A$, rather than $+A - L$, because *(ii)* the number of conclusions with particular quantity (zero in this case) is the same as the number of premises with particular quantity (zero in this case). Although we are exemplifying this logic with syllogistic inferences, this system is capable of representing relational, singular, and compound inferences with ease and clarity. Furthermore, TFL is arguably more expressive than classical first-order logic ([24], p. 172).

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
