# Peer review of "Traditional Logic and Computational Thinking"

_philosophies, doi:10.3390/philosophies6010012_

Round 1
Reviewer 1 Report
I have read with attention the manuscript entitled “Traditional logic and computational thinking” by J-Martin Castro-Manzano and submitted for publication in Philosophies.
The manuscript discusses the differences between classical and traditional logics and argues that 1) they are distinctive logics and 2) there is benefit in adopting traditional logic both to evaluate the validity of inferences in human reasonings but also to teach and address logical programming.
The formulation of arguments in philosophy being grounded in “logic”, it seems perfectly adequate to consider which logic one is talking about. For example, the principle of explosion usually causes plenty of conceptual problems to students learning classical logic as it felt somehow “unnatural” and would not reflect how they would usually use inferences in their discourse. In that respect the theme and thesis of the manuscript are clearly relevant to the readership of Philosophies.
The paper is clearly written and well presented. The thesis put forward by the author is supported by the provided references and arguments presented in the paper.
As a first minor comment, I would have liked the author to insist a bit more on the pedagogical aspects of learning a traditional logic based logic programming; maybe by giving an example in Prolog where the structure of the programme would seem overly contrived for the task at hand and usually difficult to appreciate by students.
As a second minor comment, I think that not every reader to whom this manuscript might be relevant are versed in Latin. It would be good if an English translation of the Latin expressions being used in the manuscript was provide alongside them.
Author Response
Dear Reviewer 1,
Thank you for your remarks and comments.
1. I've added an example for the sake of comparison.
2. I've added English translations where convenient.
3. I've corrected typos.
All best.
Reviewer 2 Report
The manuscript suggests that that the traditional Aristotelian logic can be useful for computational thinking. This is an important idea that can to be discussed. In this regard, the author suggests that traditional logic is not classical and that the programming emanating from traditional logic is not classical logic programming. Both suggestions is worth to consider but the manuscript leaves an impression that the notions of "classical" and "traditional" are not sufficiently defined in the text. The abstract is quite short and based on not well-defined terms "traditional" and "classic".
There are of course interpretations of Aristotelian logic which, in and of themselves, are not classical. However this presentation is not sufficient. Only if the author’s idea could be stated positively – what does non-classicality mean vis à vis the Axioms - the study will have real value.
A more extended discussion on non-classicality and traditional logic is needed to validate the main conclusions of this manuscript.
Author Response
Dear Reviewer 2,
Thank you for your remarks and comments.
1. It is true that the abstract is quite short and that it relies in the notions of "classical" and "traditional", but we believe both notions are propely defined, or rather properly developed, at page 3ff. A classical logic is defined as a system that i) uses a binary syntax (as in First Order Logic); ii) admits the explosion principle; iii) admits the positive paradox of implication; and iv) holds the reflexivity of the inference relation (i.e. p |- p) as if it were the same as the identity principle (i.e. |- p ⇒ p). Clearly, First Order Logic is classical in this sense. But traditional logic, namely syllogistic, is defined as a system that i) has its origins in Aristotelian logic; ii) uses a ternary syntax; iii) rejects explosion; iv) rejects the positive paradox of implication; and v) rejects reflexivity as an inference rule.
2. And so, following the previous definitions, from page 3 to page 5 we have tried to show that the assessments given by Boethius’ and Woods’ help us build a case to show that, indeed, there is a difference between classical First Order Logic and traditional Aristotelian logic in so far as the latter uses a term syntax, not a Fregean syntax; in so far as the notion of inference in classical logic requires reflexivity and monotonicity but does not make use of premise multiplicity, whereas traditional logic requires premise multiplicity but avoids reflexivity or monotonicity. This whole discussion serves the purpose of showing that traditional and classical are not coextensive.
All best.
Round 2
Reviewer 2 Report
The manuscript can be accepted at this stage.